# ON THE EMERGENCE OF REASONING

## ABSTRACT

Since the emergence of large reasoning models (LRMs), reasoning has often been framed as a *tree-of-thought* search, where a model traverses a discrete tree of sub-thoughts within a single chain of thought (CoT), reusing context to perform back-tracking and consistency checks. This view presupposes reasoning as a discrete search over symbolic structures. In this work, we challenge this view by proposing a new framework that conceptualizes LRM inference as *continuous optimization over an implicit energy landscape*. Here, intermediate representations correspond to positions in a high-dimensional space, and an implicit energy function encodes progress toward the solution. We motivate this perspective by showing that LRMs, unlike standard LLMs, follow smooth trajectories that make steady progress towards a solution rather than making discrete jumps. We identify *decision tokens* as checkpoints where the model *explicitly estimates energy*, and chooses to either exploit a local minimum or explore by performing larger updates, akin to basin hopping. We further demonstrate that compared to standard tokens, decision tokens operate at a slower frequency and within a distinct activation subspace, suggesting LRMs employ specialized machinery for planning and verification, analogous to the hierarchical cortical processes underlying human System 2 reasoning. Our framework unifies tree-structured reasoning and energy-based models, suggesting new directions to improve LRMs, such as improving energy estimation at decision tokens or tuning checkpoint frequencies to balance exploration and exploitation.

## 1 INTRODUCTION

Reasoning enabled via chain-of-thought (CoT) prompting requires models to generate intermediate tokens before producing a final answer significantly enhancing their performance on complex problem solving Wei et al. (2022)Yang et al. (2025). Successive tokens enhance a model's ability to integrate disparate, non co-located concepts found in training data Prystawski et al. (2023). The sequence of these tokens can also be interpreted as a discrete search trajectory over the token space Yao et al. (2023). Imposing logical structures—such as backtracking, self-validation, or planning—on this process has been shown to improve search efficiency Gandhi et al. (2025). Extensions like Yao et al. (2023) further formalize reasoning as a search over branching "thought" units, with correctness evaluated at the level of final outcomes. In parallel, other lines of work have begun to explore reasoning both as a continuous Hao et al. (2024); Zhu et al. (2025) and hybrid Yue et al. (2025) process in latent space, aiming to understand the dynamics of reasoning beyond discrete tokens.

In this work, we propose an alternative perspective on how step-by-step reasoning via sub-thoughts improves model behavior Hammoud et al. (2025). We conceptualize inference in a reasoning model as a continuous optimization process over an implicit energy landscape in latent space. Here, intermediate token representations correspond to points in a high-dimensional space, and reasoning unfolds as a smooth trajectory through these points, guided by implicit energy functions. We identify decision tokens, as key checkpoints emitted by the model when an internal progress (energy) signal crosses a threshold, thereby deciding whether to *refine* the current sub-thought or *advance* to the next. This realizes a natural exploration–exploitation tradeoff that is ubiquitous in optimization.

Furthermore, we show that decision tokens differ from standard tokens: they occur at a slower frequency and activate a distinct subspace. This aligns with recent findings on hierarchical reasoning and supports the notion that reasoning operates at multiple temporal scales. Our framework unifies perspectives from tree-structured reasoning Yao et al. (2023); Guan et al. (2025) and energy-based modeling Gladstone et al. (2025); Du et al. (2024), and it opens up new avenues for improving

large reasoning models (LRMs)—for example, by enhancing energy estimation at decision tokens or tuning the frequency of decision checkpoints to better balance exploration and exploitation.

## 2 Hierarchal Energy Optimization Formulation

We model LRM inference as optimization over *implicit* energy functions defined on a small set of hierarchical latent variables. Although we believe the model may be optimizing over many such hiearchies, we focus on two salient levels in this paper:

1. a **thought**-level hypothesis $\mathcal{T}$, which defines a compressed latent representation of a full text solution, including the logical steps and the final answer.
2. **sub-thoughts** $d$ which are sentence/paragraph-scale reasoning steps that decode to text

**Notation.** Given a question $\mathcal{Q}$ and token index $t \in \{1, \ldots, T\}$ we define:

- $\mathcal{T}_t$ denotes the current thought-level hypothesis; $\mathcal{T}^*$ is a solution-consistent fixed point.
- $d_t^{(a)}$ denotes the active sub-thought during step $t$ at refinement iteration $a$; $d_t^*$ is the finalized sub-thought for step $t$.
- At token $t$, the current thought $\mathcal{T}_t$ is updated via a function of the working set of decision tokens:
$$\mathcal{T}_{t+1} \;=\; f(\mathcal{T}_t, d_0^*, d_1^*, \ldots, d_{m(t)-1}^*, d_{m(t)}^{(0:a)})$$

**Energies.** Each of these hierarchies posit implicit, task-dependent energies that decrease as reasoning progresses:

$$E_{\mathrm{sol}}(\mathcal{Q}, \mathcal{T}) \quad \text{thought-level energy (progress toward a full solution trace),}$$
$$E_{\mathrm{sub}}(\mathcal{Q}, \mathcal{T}, d) \quad \text{sub-thought energy (progress within the current step),}$$

**Multi-frequency hypothesis.** If optimization proceeds over $\mathcal{T}$ and $d$ concurrently, we expect distinct characteristic time scales interefering with eachother.

**Relation to tree-of-thought.** Tree search presumes discrete branch expansion and selection after explicit unrolling. In our view, branch evaluation is relaxed into continuous energy descent over latent variables at multiple time-scales. From our observations, decision tokens are still emitted at key points after the model has estimated energy, and mark optimization steps at the decision token frequency.

## 3 Experiments

### 3.1 LRMs exhibit smooth trajectories

We begin by motivating our energy-based perspective of reasoning with a simple observation: LRMs exhibit *smoother hidden state dynamics* than standard LLMs. If inference is interpreted as optimization over a continuous energy function $E : \mathbb{R}^d \to \mathbb{R}$ defined on hidden states, then the evolution of the representation for token $t$ across generation steps should resemble a gradient-based update

$$h_{t+1}^{(l)} \approx h_t^{(l)} - \lambda \, \nabla_{h_t^{(l)}} E$$

where $h_t^{(l)} \in \mathbb{R}^d$ denotes the hidden state of token $t$ at layer $l$. In this view, representations evolve through small, coherent updates along a smooth trajectory, rather than abrupt jumps. This is in contrast to a discrete tree-of-thought view, where each step corresponds to a symbolic branch expansion.

To quantify smoothness, we compute the cosine similarity between successive hidden states at a fixed layer $l$:

$$\mathrm{cos\_sim}\left(h_t^{(l)}, h_{t+1}^{(l)}\right) = \frac{\langle h_t^{(l)}, h_{t+1}^{(l)} \rangle}{\|h_t^{(l)}\|_2 \, \|h_{t+1}^{(l)}\|_2}.$$

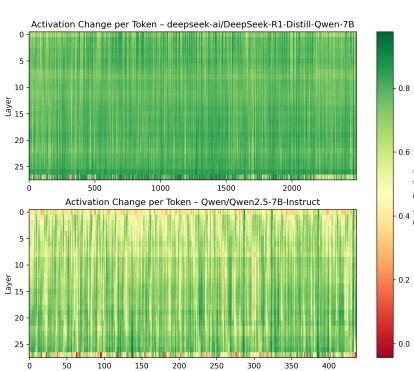 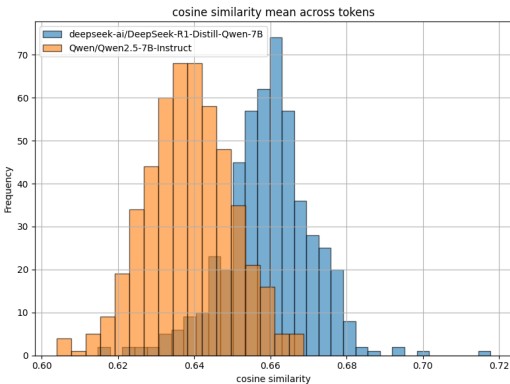

Figure 1: **Left.** Heatmap of cosine similarity between consecutive hidden states across layers (y-axis) and generation steps (x-axis) for a representative example. **Right.** Distribution of cosine similarities aggregated over all `MATH 500` examples. `DeepSeek-R1-Distill-Qwen-7B` exhibits consistently higher alignment than `Qwen2.5-7B-Instruct`, supporting the view that large reasoning models follow smoother, optimization-like trajectories through representation space.

Under the assumption that $E$ is $L$-smooth, the cosine alignment changes very slowly with $\lambda$ (see proof in Appendix A.1.1). Thus high cosine similarity indicates that consecutive representations are aligned, consistent with smooth gradient-based updates. Low similarity would suggest abrupt representational shifts, more consistent with discrete tree expansion.

Figure 7 shows a representative trajectory as well as the aggregate distribution of cosine similarities across the `MATH 500` benchmark. We find that `DeepSeek-R1-Distill-Qwen-7B` consistently exhibits higher cosine similarity across steps compared to `Qwen2.5-7B-Instruct`, supporting the hypothesis that LRMs follow smoother optimization-like trajectories.

## 3.2 HIERARCHICAL DYNAMICS

If LRMs optimize over multiple hierarchical variables, we expect their internal updates, when observed as one vector, to unfold at *multiple time scales*. To test this, we study the per-token "speed" of hidden states at the final layer $L$ (which has been shown to have the richest representations):

$$v_t^{(L)} \;=\; \left\| h_{t+1}^{(L)} - h_t^{(L)} \right\|_2$$

Figure 2 shows activation speeds for a single prompt. The raw trace (blue) is overlaid with a reconstruction from two PSD bands (orange), isolating the low and high frequency components that best explain the variance. The `DeepSeek-R1-Distill-Qwen-7B` model exhibits a clear slow oscillation in addition to a faster one, while `Qwen2.5-7B-Instruct` is dominated by a higher-frequency periodic band.

To generalize across examples, we compute the normalized cumulative power spectral density (PSD) of activation speeds over the `MATH500` benchmark. Because Qwen-Instruct's chains of thought (CoTs) are on average much shorter than DeepSeek's, we first truncate each completion to the length of the shorter trace for each example before computing spectra. As shown in panel (c), DeepSeek-Qwen consistently attributes a larger fraction of variance to longer periods, suggesting a greater role for slow dynamics. While Qwen-Instruct does show a high cumulative variance at periods beyond ∼1000 tokens, such long timescales are not meaningful as Qwen-Instruct CoTs are usually shorter than these periods.

Multiple characteristic frequencies in LRM dynamics are consistent with *hierarchical optimization*: a slower process tracking "thought-level" updates and a faster process refining sub-thoughts. In the next section we try and disentangle the activation space to isolate the modes of operation. We find that *decision tokens* align with boundaries between the hiearchies and we use this observation to gain further insight into the model machinery for dynamics at multiple frequencies.

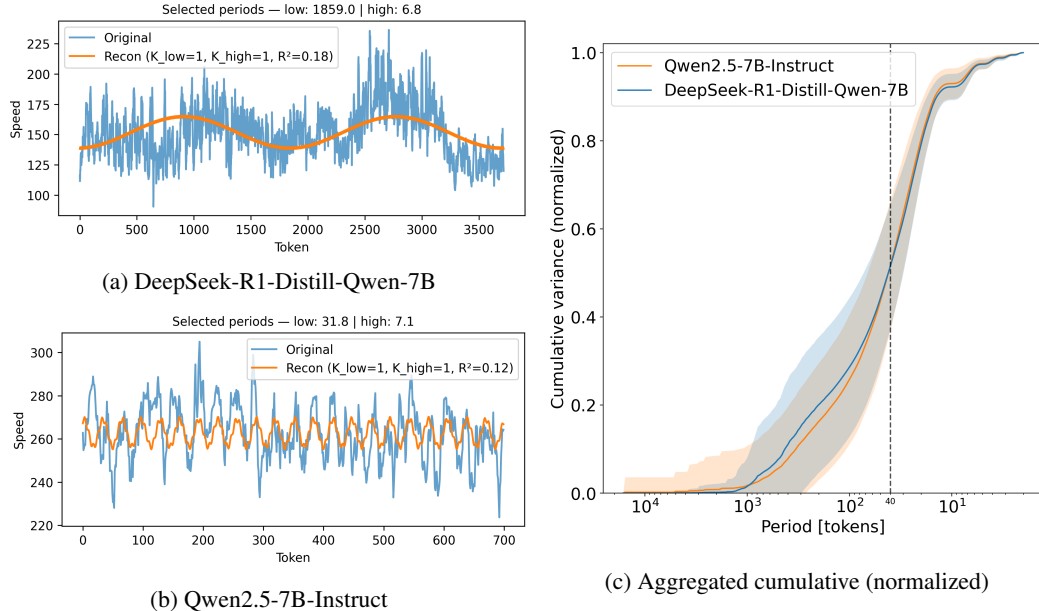

Figure 2: **(a–b)** Activation *speeds* (token-to-token norm differences at the last layer, blue) for a single prompt, shown for an LRM (a) and its LLM counterpart (b). The orange curve shows a reconstruction from two PSD bands: one low-frequency and one high-frequency component that are chosen to most explain the variance. DeepSeek exhibits a lower-frequency band with a noticeably longer period, alongside a high-frequency band of similar period to Qwen (the latter is less visible due to the scale). **(c)** Normalized cumulative variance of activation speeds vs. period (where larger from left to right = slower to faster from left to right), averaged across all `MATH500` examples. Solid lines show means; shading shows variability across examples. Higher values at long periods indicate that a larger share of variance is explained by slow dynamics. The dashed line marks the approximate transition period at 30 tokens.

## 3.3 DECISION TOKENS ARE HIERARCHAL CHECKPOINTS

We now disentangle the multi–time-scale behavior observed in 3.2 by locating (i) timesteps (tokens) and (ii) activation subspaces that index specific levels of the hierarchy. Empirically, we find that *decision tokens* mark boundaries at which the model begins decoding a new sub-thought. At these boundaries, the model (i) performs a reset in the representation used for the current sub-thought and (ii) implicitly evaluates progress (energy) before deciding whether to refine the current sub-thought or advance to the next.

Concretely, writing the active sub-thought at time $t$ as $d_t^{(a)}$ (iteration $a$ of sub-thought $t$), decision tokens coincide with either

(i) a refinement step $d_t^{(a)} \rightarrow d_t^{(a+1)}$, or

(ii) a transition $d_t^{(a)} \rightarrow d_{t+1}^{(0)}$ to begin the next sub-thought,

and occur after the model has internally estimated a progress signal $E(d_t)$ to be sufficiently low (favoring refinement) or high (favoring transition).

### 3.3.1 DECISION TOKENS REVEAL A CHARACTERISTIC SUBSPACE AND FREQUENCY

In section 3.2, we considered full final-layer activations $h_t^{(L)} \in \mathbb{R}^d$ across tokens $x_{1:T}$, where raw speed $\|h_{t+1}^{(L)} - h_t^{(L)}\|_2$ mixes multiple processes and thus obscures interpretable periodicity. Here we identify a subspace in which a single characteristic frequency dominates and within which decision tokens are distinct.

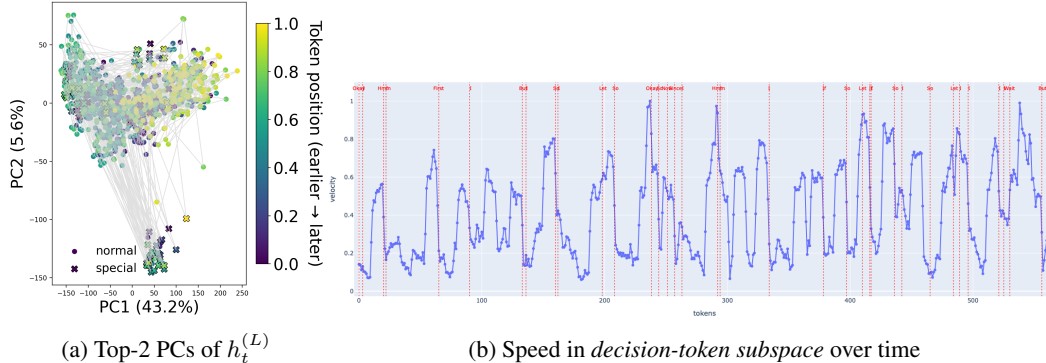

(a) Top-2 PCs of $h_t^{(L)}$        (b) Speed in *decision-token subspace* over time

Figure 3: **Decision tokens define a distinct subspace and frequency. (a)** Decision tokens (crosses) form a distinct cluster in the top principal components of the final-layer activations. **(b)** Projected speeds exhibit sawtooth signals that reset at decision tokens - consistent with sub-thought level scale dynamics.

Let $H \in \mathbb{R}^{T \times d}$ stack tokenwise activations as rows. We compute a PCA basis $W$ of $H$ and analyze the projected dynamics $z_t = W^\top h_t^{(L)}$. In Fig. 3(a), the trajectory in the top two PCs exhibits two clear clusters. One of these clusters if formed by crosses, which correspond to tokens drawn from the *decision-token* set defined in prior work (e.g., *"Wait," "But," "Then," "So,"* etc.; full list in Appendix A.2.1). This separation indicates a basis in which the model encodes decision tokens differently from intermediate tokens; we refer to the span of these PCs as the *decision-token subspace*.

Speeds calculated in this subspace ($\|z_{t+1} - z_t\|_2$), are plotted in Fig. 3(b). Here we see rapid drops in speed at decision tokens after which the speed gradually rebuilds, revealing a consistent low-frequency rhythm aligned with sub-thought boundaries. Combined with the clustering observation, this suggests that the model tracks token-level updates in the decision-token subspace until an optimization step is committed on $d_t$ (refinement or transition), after which the subspace *resets* such that the decision tokens appear to have similar representations. While one could argue that speed in this subspace merely increase until a decision token is outputted, we show in the following section that this timing is *not* periodic but causally sensitive to progress, implying the subspace carries a meaningful signal, which we refer to as an energy estimate.

### 3.3.2 CAUSAL INTERVENTION

If decision tokens indeed signal hierarchical checkpoints, their timing should depend on meaningful metrics like *progress* rather than an arbitrary fixed period for example. We test this via causal interventions that perturb progress within a sub-thought.

**Method.** For each `MATH500` prompt, we generate a baseline completion at temperature 0 and identify decision tokens $\{d_t\}_{t=1}^T$. For a uniformly sampled decision index $t$, we measure the baseline gap $k$ (tokens from $d_t$ to the next decision token $d_{t+1}$). We then restart generation from the context starting at $d_t$ but sample completions at a high temperature to inject off-policy perturbations. Finally we record $k'$, the number of tokens until a decision token $d'_{t+1}$ becomes the most probable token. We then record the proportion of examples where $k' < k$.

**Result.** Over 500 traces, we observe a decision token is emitted **earlier** when progress degrades. This result indicates that the emission of a decision token is *causally linked* to a progress signal—consistent with an internal estimate $E(d_t)$.

If reasoning were a strict *tree-of-sub-thoughts* search, a decision point would be emitted only after fully unrolling a sub-thought to select among branches. Instead, we observe preemptive boundary insertion as soon as progress deteriorates.

| temp | p $\pm$ SE |
|------|------------|
| 0.5  | $0.604 \pm 0.0219$ |
| 2.0  | $0.69 \pm 0.0207$ |
| 3.0  | $0.706 \pm 0.0206$ |

Table 1: Temperature and Error

Moreover, sweeping the sampling temperature yields a positive relationship between temperature and the advancement $k - k'$.

The noisier (and thus less progressive) the continuation, the earlier the boundary appears. This behavior reflects a graded progress signal rather than a binary dead-end flag, and suggests that timing is driven by hidden-state dynamics (e.g., the decision-token subspace) rather than by the lexical identity of the decision tokens themselves.

We also look at examples A.2.4 where the the model reaches a dead end $d_t^0$ and re-optimizes $d_t^1$. We then intervene by giving it the correct $d_t^*$, and find the model more quickly outputs a decision token and moves to $d_{t+1}^0$. As an orthogonal check, when we provide corrective hintsA.2.4 that align with the ground-truth sub-thought $d_t^*$, the next boundary occurs *sooner* as the model exits refinement and initiates the next sub-thought, again matching the energy-checkpoint view. In particular, low estimated progress triggers refinement ($d_t^{(a)} \to d_t^{(a+1)}$) and earlier resets; high progress favors advancing to $d_{t+1}^{(0)}$.

### 3.4 OPTIMIZATION OVER HIERARCHIES

Thus far we have argued that LRMs behave as if optimizing over an implicit energy functions, but we have not analyzed how they traverse such spaces. In this section, we propose estimates for energies associated with multiple hierarchies and observe the models trajectories over these landscapes. At the thought level, the objective is to assemble a complete *solution* via a chain-of-thought. At a finer scale, the objective is to optimize sub-thoughts to steer towards producing the final *answer*. We therefore define two progress measures:

(i) a *solution energy* that reflects the model's progress towards producing a particular full solution trace, and (ii) an *answer energy* that reflects progress toward emitting the final boxed answer. Practically, we track these over time by *forcing* (appending) a candidate solution/answer at every prefix of the generated CoT and measuring the (log-)probability the model assigns to that forced continuation. In this way, we are actually measuring and plotting the inverse energy in the following sections. [1]

#### 3.4.1 THOUGHT OPTIMIZATION

We first estimate progress toward a *full solution trace*. Let $\mathcal{T}_t$ denote the current thought-state at token $t$ and let $S$ be a fixed solution trace (a sequence of tokens) that captures one correct line of reasoning. To probe $E(\mathcal{T}_t)$, we append a delimiter ("`Wait, `") followed by $S$ to each CoT prefix and compute the summed log-probability assigned to $S$ by the model; higher values indicate the model is closer to reproducing that particular solution. We only consider this particular solution for now as we know that if the model is indeed optimizing for a solution, the solution it outputted is a likely candidate.

For `DeepSeek-R1-Distill-Qwen-7B`, we take the model's own final solution (after the `</think>` tag) as $S$, and also construct paraphrases by rewording and re-explaining the same logic. For `Qwen2.5-7B-Instruct`, we use its (typically shorter) final solution as $S$ and similarly generate paraphrases. Figure 4 plots the resulting *solution energy* curves for the original $S$, reworded $S$, and re-explained $S$. While both models show a gradual rise for the *exact* solution they produced, `Qwen-Instruct` becomes increasingly unlikely to produce paraphrases of its own solution. In contrast, `DeepSeek` exhibits invariant progress curves across paraphrases (note the comparable logit scales), suggesting it maintains and optimizes a *compressed latent representation* of the solution.

#### 3.4.2 SUB-THOUGHT OPTIMIZATION

We next estimate a higher-frequency objective defined by the models ability to produce the final answer. Let $A^\star$ be the ground-truth answer string. At each CoT prefix, we append the template "`\nTherefore the answer is \boxed{`$A^\star$`}`" and compute the summed log-probability over the answer tokens. We refer to this quantity as the *answer energy* (again, higher log-probability $\Leftrightarrow$ lower energy / greater progress). We can think of such a probability as the probability of decoding the final subthought representation $d_T$ into the answer.

---

[1] For plots we sum token log-probabilities of the forced span.

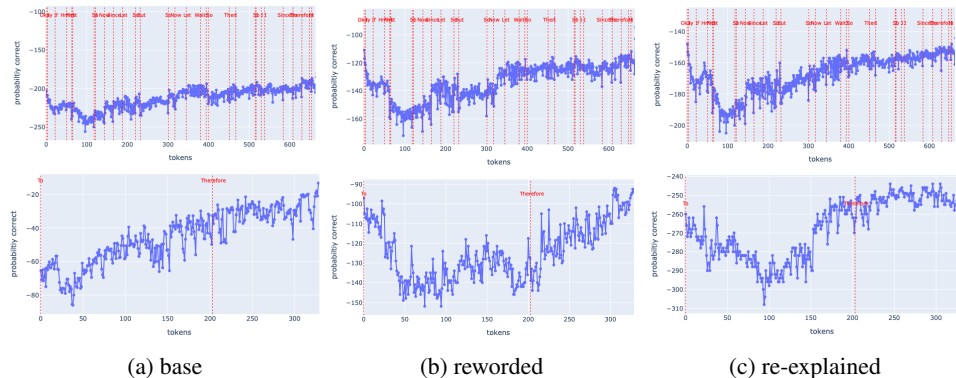

(a) base       (b) reworded       (c) re-explained

Figure 4: **Inverse *solution energy* (thought-level progress) for LRM vs. LLM.** Top: `DeepSeek-R1-Distill-Qwen-7B`. Bottom: `Qwen2.5-7B-Instruct`. Each panel shows the (summed) log-probability of a forced full solution trace appended to each CoT prefix. Left: original solution; middle: reworded; right: re-explained. `DeepSeek` exhibits progress curves that are largely invariant to paraphrasing, consistent with the notion that the model is steadily building and optimizing a latent representation of the solution

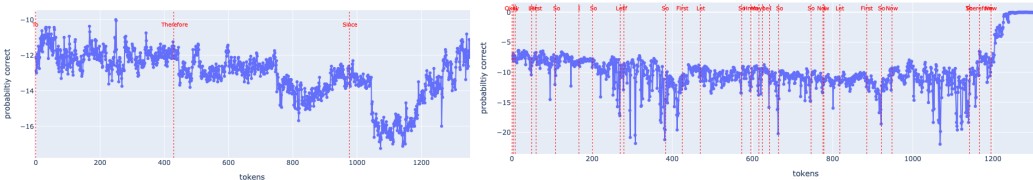

Figure 5: **Inverse *answer energy* (sub-thought-level progress) for `Qwen-Instruct` (left) vs. `DeepSeek` (right).** Curves show the (summed) log-probability of a forced ground-truth answer appended to each CoT prefix. `DeepSeek` exhibits smooth, structured growth with discrete jumps at decision tokens (dashed red lines); `Qwen-Instruct` displays noisier dynamics lacking clear hierarchical structure.

Figure 5 compares answer energy trajectories for a difficult problem that `DeepSeek` solves and `Qwen-Instruct` misses (the `DeepSeek` trace is truncated to match `Qwen`'s length for comparability). `Qwen-Instruct` shows high-variance and no interpretable structured search over the space. While `DeepSeek` exhibits a smoother trajectory, barring key jumps at decision tokens (vertical dashed lines), consistent with hierarchical optimization observations in 3.3.1. This smooth optimizes of an energy landscape, indicates that $d_T$ may be updated along with $d_{1:T-1}$ during inference. Later in the trace, after earlier sub-thoughts $d_{1:T-1}$ optimization stabilizes, the answer probabilty rises steadily towards 1, indicating that the final sub-thought $d_T$ is refined last to commit to the boxed answer.

Finally, Fig. 6 contrasts the ground-truth answer with incorrect alternatives: probabilities for the correct answer jump *up* at decision tokens, while those for incorrect answers jump *down*. This divergence suggests that the decision tokens may have some representation, $d_T$ that aligns with the ground truth answer. This observation aligns with prior observations of elevated mutual information between decision token representations and solution features Qian et al. (2025).

## 4 RELATED WORKS

### 4.1 ENERGY BASED MODELS

Existing energy-based reasoning approaches model problem solving as iterative optimization on learned energy landscapes over a token level configuration space. Du et al. (2022) treats reasoning as adaptive energy minimization, where local minimas in energy provide natural stopping points

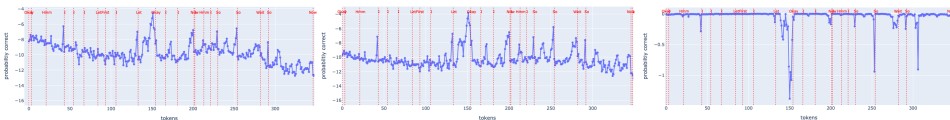

Figure 6: **Counterfactual answer energy separates correct from incorrect alternatives.** At each CoT prefix we force "\nTherefore the answer is \boxed{a}" and plot the summed log-probability over the answer tokens (y-axis) vs. token index (x-axis). Panels show two incorrect candidates ($a = 7$, $a = 8$) and the ground-truth ($a = 9$). Vertical dashed lines mark decision tokens. Probabilities *decrease* at decision tokens for incorrect answers (left, middle) but *increase* for the correct answer (right). This divergence is consistent with the hypothesis the latent $d_t$ tracks progress toward the true solution.

for output. In a follow-up paper, Du et al. (2024) reformulates reasoning as an energy-based optimization where energy functions are learned over input/output pairs and models solve for outputs by iteratively minimizing energy, using a sequence of *annealed energy landscapes*. Gladstone et al. (2025) extends this idea to large architectures, assigning energies to candidate predictions and refining them via gradient descent, yielding stronger out-of-distribution generalization and scalable reasoning. Together, these works show that energy-based formulations support structured, controllable reasoning beyond standard CoT. In our work however, we look at energy optimizations beyond just the high frequency optimizations of token completions, and consider other latent optimizations occuring at different scales.

## 4.2 LATENT SPACE REASONING

Other works have motivated that exploring latent reasoning dynamics in LRMs is critical to understand reasoning behavior beyond tokens. Geiping et al. (2025) introduces a model architecture that lets the model iterate a recurrent latent block at inference time (unrolling to arbitrary depth) rather than generating longer CoT. It shows that this form of compute scaling in latent space can dramatically improve performance without requiring longer context windows or specialized CoT-style training data.Lee et al. (2024) shows that neural networks can perform multiple steps of mathematical reasoning within a fixed-dimensional latent space. Finally Wang et al. (2025a) introduces a Hierarchical Reasoning Model with high-level and low-level recurrent modules, achieving strong reasoning performance without chain-of-thought supervision. Our work, extends these perspectives of latent trajectories at multiple frequencies to CoT reasoning by arguing that even a standard LLM is capable of tracking and optimizing latent representations in activation space.

## 4.3 COT TOKENS THEMSELVES DON'T MATTER

Recent work increasingly suggests that the reasoning abilities of language models rely more on latent computation and internal representations than on the specific tokens emitted during inference. Pfau et al. (2024) show that models can perform comparably well when intermediate reasoning steps are replaced with filler tokens, implying that additional tokens act primarily as compute scaffolding rather than semantically meaningful content. Turpin et al. (2023) find that chain-of-thought (CoT) rationales are often unfaithful to the model's actual decision process, functioning more as post hoc justifications. Approaches like Coconut Hao et al. (2024), CODI Shen et al. (2025), and System-1.5 Reasoning Wang et al. (2025b) further demonstrate that continuous latent-space reasoning can match or outperform explicit CoT while emitting fewer or no intermediate tokens. These findings also align with interpretability work in Zhang & Nanda (2024) which shows that critical computation often occurs in hidden states, regardless of what the model verbalizes. Collectively, this body of work indicates that token-level reasoning may be more superficial than previously assumed, with the true reasoning residing in the model's internal activation dynamics.

## 5 LIMITATIONS AND FUTURE WORK

**Limitations.** Our analysis provides a promising first step towards establishing reasoning as continuous energy descent. However, our observations are limited to experiments run on

`DeepSeek-R1-Distill-Qwen-7B` and `Qwen2.5-7B-Instruct` over examples mostly taken from the `MATH 500`. We hope to expand the scale of our experiments and validate our hypothesis across other model families and scale.

**Future Work.** We plan to further substantiate our claims by looking at the models attention at decision tokens and within the decision-token subspace (Section 3.3.1) to test whether sub-thought content is explicitly represented and used for decoding.

Additionally we plan to run causal masking experiments to show that these subspaces are necessary to decode meaningful progress in an energy landscape. We also hope to show by attention patching that we can improve or degrade latent representations and decode solutions faster.

Beyond the thought and sub-thought bands, we will search for additional characteristic frequencies of reasoning. Finally, we aim to train energy-based transformers with explicit multi-level progress heads and checkpoint regularizers, aligning training objectives with hierarchical energies to improve reliability and sample efficiency.

**Conclusion** In this work, we introduced a new framework that reformulates inference in Large Reasoning Models (LRMs) as continuous optimization over an implicit energy landscape. In this view, intermediate representations correspond to positions in a high-dimensional latent space, and reasoning unfolds as a smooth trajectory shaped by descent along an implicit energy function encoding progress toward a solution. This energy-based perspective offers a unifying account of emergent reasoning, bridging symbolic tree-structured search with continuous optimization.

Our empirical analysis revealed that LRMs—unlike standard LLMs—exhibit smooth, coherent latent trajectories consistent with gradient-based optimization. We identified hierarchical temporal dynamics, where slower "thought-level" decisions are interleaved with faster "sub-thought" refinements. Central to this hierarchy are decision tokens, which serve as implicit checkpoints in a distinct activation subspace, marking transitions between exploration and exploitation. Causal interventions further showed that the timing of these tokens reflects an implicit latent progress signal. By defining proxies for solution energy and answer energy, we demonstrated that LRMs make consistent, paraphrase-invariant progress toward solutions with decision tokens marking divergence points between correct and incorrect reasoning paths.

Our framework opens new directions for building more robust, efficient, and interpretable models by shaping their internal energy trajectories, and optimizing tradeoffs between exploration and convergence.

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

# A APPENDIX

## A.1 MATH PROOFS

### A.1.1 COSINE ALIGNMENT PROOF

Here we provide a formal derivation supporting the claim in Section 3.1.

Let $E : \mathbb{R}^d \to \mathbb{R}$ be an $L$-smooth function, i.e.

$$\|\nabla E(x) - \nabla E(y)\| \leq L\|x - y\| \qquad \forall x, y.$$

Furthermore, the gradient-based update rule is given by

$$h_{t+1} = h_t - \eta \nabla E(h_t),$$

where $\eta > 0$ is the step size.

Then the cosine similarity between $\nabla E(h_{t+1})$ and $\nabla E(h_t)$ is high.

*Proof.* We define $g_t := \nabla E(h_t)$ and $g_{t+1} := \nabla E(h_{t+1})$.

Proof: Using the definition of smoothness we can write

$$\|g_{t+1} - g_t\| \leq L\|h_{t+1} - h_t\|. \tag{1}$$

Applying the update rule to equation 1 we get

$$\|g_{t+1} - g_t\| \leq L\| - \eta g_t\| \leq \eta L \|g_t\|. \tag{2}$$

Now consider the unit vectors $\hat{g}_t := g_t/\|g_t\|$ and $\hat{g}_{t+1} := g_{t+1}/\|g_{t+1}\|$. Then

$$\begin{aligned}
\|\hat{g}_{t+1} - \hat{g}_t\|^2 &= \|\hat{g}_{t+1}\|^2 + \|\hat{g}_t\|^2 - 2\|\hat{g}_{t+1}\|\|\hat{g}_t\|\cos\theta \\
&= 2(1 - \cos\theta),
\end{aligned} \tag{3}$$

Now,

$$\begin{aligned}
\|\hat{g}_{t+1} - \hat{g}_t\| &= \left\| \frac{g_{t+1}}{\|g_{t+1}\|} - \frac{g_t}{\|g_t\|} \right\| \\
&= \left\| \frac{g_{t+1}\|g_t\| - g_t\|g_{t+1}\|}{\|g_{t+1}g_t\|} \right\|
\end{aligned} \tag{4}$$

We now upper bound the numerator and lower bound the denominator in (4).

$$\begin{aligned}
\|g_{t+1}\|g_t\| - g_t\|g_{t+1}\|\| &\leq \|(g_{t+1} - g_t)\|g_t\| - g_t\|g_{t+1} - g_t\|\| \\
&\leq \|(g_{t+1} - g_t)\|\|g_t\| + \|g_t\|\|g_{t+1} - g_t\|\| \\
&\leq 2\|g_{t+1} - g_t\|\|g_t\|
\end{aligned} \tag{5}$$

Substitute the bound equation 2 into equation 5 to get

$$\|g_{t+1} - g_t\|^2 \leq 2(\eta L)\|g_t\|^2. \tag{3}$$

For the denominator use the reverse triangle inequality:

$$\|\|g_{t+1}\| - \|g_t\|\| \leq \|g_{t+1} - g_t\| \qquad \text{(Reverse triangle inequality)}$$
$$\pm(\|g_{t+1}\| - \|g_t\|) \leq \|g_{t+1} - g_t\| \tag{7}$$
$$\|g_{t+1}\| - \|g_t\| \leq \|g_{t+1} - g_t\|$$

Thus

$$\|g_{t+1}\| \geq \|g_t\| - \|g_{t+1} - g_t\|$$
$$\geq (1 - \eta L)\|g_t\|, \tag{8}$$

Thus the denominator can be lower bounded by

$$\|g_t\| \|g_{t+1}\| \geq (1 - \eta L)\|g_t\|^2. \tag{4}$$

Substituting equation 3 and equation 4 into equation 4 yields

$$\|\hat{g}_{t+1} - \hat{g}_t\| \leq \frac{2\eta L\|g_t\|^2}{2(1 - \eta L)\|g_t\|^2} = \frac{2\eta L}{1 - \eta L}. \tag{5}$$

Substituting equation 5 into equation 3 we get

$$2(1 - \cos\theta) \leq 4\left(\frac{L\eta}{1 - L\eta}\right)^2$$

$$\implies \cos\theta \geq 1 - 2\left(\frac{L\eta}{1 - L\eta}\right)^2$$

Therefore, for sufficiently small step-size $\eta$ (so that $1 - \eta L > 0$), the quantity $\cos\theta$ is close to 1 which proves that the gradients $g_t$ and $g_{t+1}$ are aligned. $\qquad\square$

## A.2 EXPERIMENTAL DETAILS

All experiments are run with greedy sampling unless otherwise specified.

### A.2.1 REASONING TRACES

We define the following as chosen decision tokens and split thoughts into subthoughts, as done in Hammoud et al. (2025)

```
{"So", "Let", "Hmm", "I", "Okay", "First", "Wait", "But", "Now",
"Then", "Since", "Therefore", "If", "Maybe", "To"}
```

For all reasoning traces, we use the following system prompt to generate rollouts.

```
system_prompt = "Please reason step by step, and put your final
answer within \boxed{}."
```

**Frequency band reconstruction** We choose one lower frequency, and one higher frequency PSD. The low and high frequency points are decided by a threshold boundary set at a frequency of 0.01. We then choose the largest PSD peak within the categories. The speeds are plotted with a rolling average of 10 to denoise high frequencies for better visualization.

**Normalized Cumulative Variance** To compare across examples and models, we compute the cumulative PSD normalized by its total variance. The normalized cumulative curve thus reaches 1 and indicates the fraction of total variance explained by oscillations with period greater than or equal to $T$. Aggregating across all 500 MATH examples, we report mean and standard deviation bands for each model on a common frequency grid.

### A.2.2 SOLUTION ENERGY

Here is the example, the reworded and the re-explained text: lets plot some more examples of solution energy here

### A.2.3 ANSWER ENERGY

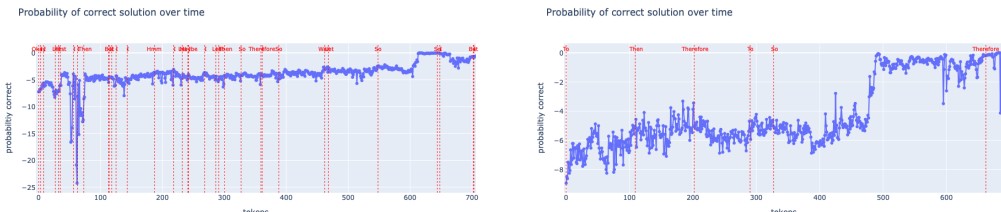

Figure 7: **answer energy for LLM vs LRM**. Here we show an example of the probablity of forced ground truth answer two models for a question they both answer correctly. For comparison deepseek trace has been truncated to match qwens length. Over this period it is evident how even what we call the sub-thought latent $d_T$ evolves more smoothly and with more structure. you can also see clear spikes at decision tokens marked as dashed red vertical lines.

### A.2.4 CAUSAL INTERVENTION

**Example 2.4.1**

---

$d_{t-1}$: Okay, so I have this geometric sequence: 125/9, 25/3, 5, 3, and so on. I need to find the eighth term. Hmm, let me think about how geometric sequences work.

First, I remember that in a geometric sequence, each term is found by multiplying the previous term by a common ratio, which is usually denoted as 'r'. So, if I can figure out what 'r' is, I can find any term in the sequence, including the eighth one.

Let me write down the terms again to see the pattern:

1st term: 125/9

2nd term: 25/3

3rd term: 5

4th term: 3 So, to find the common ratio, I can divide the second term by the first term. Let me calculate that:

$r = \frac{25/3}{125/9}$

Dividing fractions is the same as multiplying by the reciprocal, so:

$r = \frac{25}{3} \cdot \frac{9}{125}$ First, I remember that in a geometric sequence, each term is found by multiplying the previous term by a common ratio, $r$. So, to find $r$, I can divide the second term by the first term. Let me calculate that: $(25/3)$ divided by $(125/9)$.

---

$d_t^0$: Wait, let me check that again. 25/3 divided by 125/9 is $\left(\frac{25}{3}\right) \cdot \left(\frac{9}{125}\right)$. 25 times 9 is 225, and 3 times 125 is 375. So, $\frac{225}{375}$. Simplifying that, both numerator and denominator can be divided by 75. $225 \div 75 = 3$, and $375 \div 75 = 5$. So, yes, $r = 3/5$. That seems right.

---

$d_t^*$: Let me verify with the next term. If I take $25/3$ and multiply by $3/5$, I should get 5. Let's see: $\left(\frac{25}{3}\right) \cdot \left(\frac{3}{5}\right) = \frac{25 \cdot 3}{3 \cdot 5} = \frac{75}{15} = 5$. Perfect, that works.

---

$d_{t+1}$: So, the common ratio $r$ is $3/5$. Now, the formula for the nth term of a geometric sequence is $a_n = a_1 \cdot r^{n-1}$. Here, $a_1$ is 125/9, and $r$ is 3/5. I need to find the eighth term, so $n = 8$.

---

**Example 2.4.2**

---

$d_{t-1}$: Okay, so Frank took this aptitude test with 100 questions. Each correct answer gives him 5 points, each incorrect one deducts 2 points, and leaving a question blank gives him nothing. He answered 80 questions and scored 232 points. I need to figure out how many he got right.

---

$d_t^0$: First, there are 100 questions in total. Frank answered 80, so that means he left 20 blank. .

---

$d_t^1$: So, we know that: $c + i = 80$ Because he answered 80 questions in total

$d_t^*$: Now the total score is $5c - 2i$

$d_{t+1}$ : Which is 232. So, we have two equations:

### Example 2.4.3

$d_{t-1}$: Okay, so I need to compute 99 squared plus 99 plus 1 in my head. Hmm, that seems a bit tricky at first glance, but maybe there's a clever way to break it down. Let me think about how to approach this.

$d_t^0$: Wait, maybe I can think of it as $(99+1)^2$ - something. Let's see: $(99+1)^2 = 100^2 = 10000$. But $99^2 + 99 + 1$ is $99^2 + 99 + 1$. Let me compute $(99 + 1)^2$: that's $99^2 + 2 \cdot 99 + 1$. So, $99^2 + 2 \cdot 99 + 1 = 10000$.

$d_t^*$: Therefore, $99^2 + 99 + 1$ is equal to $(99^2 + 2 \cdot 99 + 1) - 99 = 10000 - 99 = 9901$. Oh, that's a neat trick!

$d_{t+1}$ : So, by recognizing that $(99 + 1)^2$ is $100^2 = 10000$, and then subtracting 99, I get the result.

### Example 2.4.4

$d_{t-1}$: Okay, so I have this problem here: A point $(x, y)$ is randomly selected such that $0 \leq x \leq 8$ and $0 \leq y \leq 4$. I need to find the probability that $x + y \leq 4$. Hmm, let me think about how to approach this.

$d_t^0$ First, I know that when dealing with probabilities involving areas, especially with uniform distributions, the probability is usually the ratio of the favorable area to the total area.

$d_t^*$ So, in this case, the total area where the point can be selected is a rectangle with length 8 and width 4. Let me calculate that.

**Hint:** the area of the triangle based on $x + y \leq 4$ is ...

$d_{t+1}$: is 8 so the probability is $8/32$.

### Example 2.4.5

$d_{t-1}$: Okay, so I have this problem where I need to find the matrix A multiplied by another vector, specifically A times (-13, 3, 4). I know that A is a matrix that transforms two given vectors into two other vectors. Let me write down what I know: First, A multiplied by the vector (3, 1, 0) gives (2, 4, -3). So, mathematically, that's A * $[3, 1, 0]^T = [2, 4, -3]^T$ Second, A multiplied by (-5, 2, 2) gives (0, -5, -5). So, A * $[-5, 2, 2]^T = [0, -5, -5]^T$
And I need to find A multiplied by (-13, 3, 4). Hmm, okay

$d_t^*$: Hmm, but I need to express (-13, 3, 4) as a combination of these two vectors.

**Hint:** Let me check if they are linearly independent.

$d_{t+1}$: If I set up the equation: c1*(0, -5, -5) + c2*(something) = (-13, 3, 4)

