# OpenReview forum: "On the Emergence of Reasoning"
_ICLR.cc/2026/Conference — ICLR 2026 Conference Withdrawn Submission_

### Official Review · Reviewer_pe4G · 2025-10-21

**Soundness:** 2
**Presentation:** 3
**Contribution:** 2
**Rating:** 4
**Confidence:** 3

**Summary:**

The paper presents a novel hypothesis that reframes reasoning in large models as continuous optimization over a latent energy landscape, rather than as a discrete symbolic process. The authors introduce a hierarchical energy optimization formulation, distinguishing between fast sub-thought-level and slower thought-level refinements. Empirically, the paper compares a reasoning model to a baseline LLM on the MATH 500 benchmark. The authors observe that LRMs exhibit smoother trajectories and distinct patterns for specific ``decision tokens".

**Strengths:**

The paper proposes a novel and interesting hypothesis, arguing that reasoning can be seen as continuous optimization over a latent energy landscape. Rather than viewing reasoning as a discrete tree search, the authors argue that models evolve smoothly, progressively minimizing an implicit energy function. The paper lays important groundwork for future improvements in reasoning models.

**Weaknesses:**

1. The evidence that ``decision tokens" act as reasoning checkpoints is suggestive but not conclusive. Many of these tokens frequently occur at the beginning of the sentence, so the observed activation resets could simply reflect sentence boundaries rather than reasoning transitions. The paper does not seem to control for this confound.

2. The experiments seem rather limited, since the paper only compares one reasoning model and a single baseline LLM on a single dataset. I believe this is not sufficient to draw any meaningful conclusions since differences between the two could just be model- or dataset-specific rather than a more general phenomenon.

3. I am not fully convinced that there is enough evidence to suggest that high cosine similarity and smooth trajectories are indicative of better reasoning. These metrics may simply reflect general coherence rather than improved reasoning.

4. The energy functions are not formally defined or directly optimized. Instead, the authors interpret hidden-state smoothness as a proxy, without providing concrete evidence that such an underlying energy function actually exists.

**Questions:**

1. Have you compared decision tokens to other frequent sentence starters to rule out syntactic confounds?

2. How could future models explicitly incorporate or optimize over the energy landscape?

---

### Official Review · Reviewer_GUoe · 2025-10-30

**Soundness:** 2
**Presentation:** 3
**Contribution:** 2
**Rating:** 4
**Confidence:** 4

**Summary:**

This paper presents a conceptual and empirical investigation into how reasoning may emerge in LRM. Rather than performing explicit “tree-of-thought” exploration, the paper shows that LRMs tend to exhibit continuous energy minimization dynamics in their hidden-state trajectories.

**Strengths:**

- The paper offers a connection beteen  reasoning, optimization, and representation geometry shoing  why CoT and ToT behavior emerges with scale.

- the paper provides efficent and easy to implement probing thechniques.

- the paper is well motivated and well structured.

**Weaknesses:**

- the link between gradient alignment (\nabla E_t, \nabla E_{t+1}) and state-state cosine (h_t, h_{t+1}) is not clear.  how this link to  residual connections and layer normalization?

- it is not clear how do the energy probe results behave under contrastive or randomized continuations?

- The decision-token set seems corrolated with style and ponctuation. It is not clear to what extent this affect PCA?

- Evaluation is quite limitted : two 7B models and one domain (MATH500).

**Questions:**

see Weaknesses

---

### Official Review · Reviewer_xAh9 · 2025-10-31

**Soundness:** 3
**Presentation:** 2
**Contribution:** 4
**Rating:** 6
**Confidence:** 4

**Summary:**

The authors propose to interpret LRMs reasoning as implicit energy minimization. The authors broadly support this claim with empirical experiments comparing an LRM and a base LLM, demonstrating the LRM has dynamics more similar to EBMs across hidden states, hierarchical dynamics, and "energy" measurements via probabilities.

**Strengths:**

- The perspective of LRMs as minimizing an implicit energy function is interesting, novel, and useful for the community. I find this paper fascinating and, regardless of whether or not the evidence is very strong or clear, useful insight for people. Despite listing many weaknesses, I see them as smaller compared to this main strength.
- The findings may have use for the mechanistic interpretability community.
- The idea to force ground truth trajectories and use logprobs as inverse energies is clever and useful.

**Weaknesses:**

- In Figure 1, the evidence for reasoning models having much smoother implicit gradient dynamics than instruct models is somewhat weak, as the reasoning models still don't have super high cosine similarity across consecutive and tend to have "discrete" jumps (just less of them).
- The overall strength of the evidence is mostly based on single models, data points, and plots, it would be nice to confirm the significance of the results across models and more data as well as with hypothesis testing.
- Just because the LRMs exhibits dynamics more similar to implicit EBMs than LLMs, it doesn't mean the LRMs are truly minimizing an implicit energy function.
- I found the results in Figures 4 and 5 to be very weak. I would have liked to see more text pieces measured (instead of just 1) as well as potentially larger differences between models in 4 and a clearer "energy descent" in Figure 5 for deepseek.
- Overall, the largest weaknesses primarily have to do with weaker evidence for the general claims of the paper. I would be more confident in the paper if experiments were made to be more significant via testing with more data and models.

**Questions:**

- Are the dynamics inferred to be optimizing a single global implicit energy function or several hierarchical energy functions? If there are several hierarchical energy functions, the decision token observations make sense, otherwise, wouldn't they contradict the continuous optimization claim? Or, is the claim that gradient steps are taken at each decision token step?
- I was a bit confused by the causal intervention results and the evidence with them as well as why the experiments were conducted the way they were---I think that could be explained better.

---

### Official Review · Reviewer_tiKa · 2025-10-31

**Soundness:** 2
**Presentation:** 1
**Contribution:** 2
**Rating:** 2
**Confidence:** 4

**Summary:**

This paper proposes a novel framework for understanding inference in Large Reasoning Models as continuous optimization over implicit energy landscapes, rather than discrete tree-of-thought search.  They argue that intermediate representations correspond to positions in high-dimensional space, with reasoning unfolding as smooth trajectories guided by energy functions. They identify "decision tokens" as checkpoints where models estimate energy and balance between exploitation and exploration.

**Strengths:**

1. The energy-based optimization perspective on LRM reasoning is creative and offers a fresh lens for understanding chain-of-thought processes beyond discrete symbolic search.
2. Their framework attempts to unify tree-structured reasoning with energy-based models, which is an ambitious theoretical contribution.

**Weaknesses:**

1. The paper does not appear ready for submission. The citation style, figure formatting, and reference section contain multiple errors and inconsistencies that prevent proper compilation. For example, Figure 4 includes severe text overlap, making it unreadable, and the figure itself is not properly embedded as a standard PDF image.

2. The paper evaluates only one model pair: DeepSeek-R1-Distill-Qwen-7B and Qwen2.5-7B-Instruct. This is insufficient to support broad claims about reasoning dynamics. Moreover, MATH500 represents a narrow domain focused on mathematical reasoning. It remains unclear how the findings would generalize to other reasoning types, such as commonsense, logical, or causal reasoning.

3. Their so-called ``energy functions" are never formally defined, making this concept difficult to understand. The paper merely asserts their implicit nature without explaining their mathematical form, theoretical foundation, or link to model training objectives. It is therefore unclear what qualifies these quantities as energy functions rather than arbitrary progress or loss metrics.

4. The paper provides no mathematical derivation or proof supporting the claim that autoregressive LLMs perform energy minimization during reasoning.

**Questions:**

1. The paper discusses reasoning and uncertainty from an energy perspective, but it is unclear whether a knowledge entropy-based analysis is necessary or complementary to the presented framework. I suggest the authors consider and discuss connections to related work, such as Kim et al. (2025) [1].

2. In Table 1, the authors test temperature values of 0.5, 2.0, and 3.0, but do not explain this selection. Typically, LLMs are evaluated with temperatures around 0.6 and 0.7. Could the authors justify why these specific (and somewhat extreme) values were chosen and discuss whether the observed trends hold under more standard temperature settings?

3. The paper does not clearly describe how decision tokens are identified. Are these detected through string matching, token ID mapping, or another automated method? A more detailed explanation would help readers understand and reproduce the analysis.


[1] Kim, Jiyeon, et al. "Knowledge Entropy Decay during Language Model Pretraining Hinders New Knowledge Acquisition." The Thirteenth International Conference on Learning Representations.

---

### Note · Authors · 2025-11-27

I have read and agree with the venue's withdrawal policy on behalf of myself and my co-authors.